# Neurophysiological Effects of Withdrawal from Acute Overused Medications in Chronic Migraine with Medication-Overuse Headache

**DOI:** 10.3390/jcm13237491

**Published:** 2024-12-09

**Authors:** Gabriele Sebastianelli, Francesco Casillo, Chiara Abagnale, Antonio Di Renzo, Lucia Ziccardi, Vincenzo Parisi, Cherubino Di Lorenzo, Mariano Serrao, Gianluca Coppola

**Affiliations:** 1Department of Medico-Surgical Sciences and Biotechnologies, Polo Pontino, Sapienza University of Rome, 04100 Latina, Italy; gabriele.sebastianelli@uniroma1.it (G.S.); francesco.casillo@uniroma1.it (F.C.); chiara.abagnale@uniroma1.it (C.A.); cherub@inwind.it (C.D.L.); mariano.serrao@uniroma1.it (M.S.); gianluca.coppola@uniroma1.it (G.C.); 2IRCCS—Fondazione Bietti, 00184 Rome, Italy; antoniomp777@hotmail.it (A.D.R.); lucia.ziccardi@fondazionebietti.it (L.Z.)

**Keywords:** medication overuse, sensitization, somatosensory system, thalamus, lateral inhibition, chronic migraine, medication withdrawal

## Abstract

**Background/Objectives**: Chronic migraine with medication-overuse headache (CM-MOH) is neurophysiologically characterized by increased cortical excitability with sensitization at both the thalamocortical and the cortical levels. It is unclear whether the increased cortical excitability could be reverted by medication withdrawal (i.e., brain state) or whether it is a brain trait of individuals predisposed to medication overuse. In this study, we aim to investigate whether withdrawal from overused drugs can influence and restore these neurophysiological abnormalities. **Methods**: Somatosensory evoked potentials (SSEPs) were elicited by electrical stimulation of the median nerve (M), the ulnar nerve (U), and the simultaneous stimulation of both nerves (MU) in 14 patients with CM-MOH before (T_0_) and after (T_1_) a three-week withdrawal protocol and, for comparison, in 14 healthy volunteers (HVs) of a comparable age distribution. We measured the level of thalamocortical (pre-HFO) and cortical activation (post-HFO) by analyzing the high-frequency oscillations (HFOs) embedded in parietal N20 median SSEPs. Furthermore, we calculated the habituation and the degree of cortical lateral inhibition (dLI) of N20-P25 low-frequency SSEPs. **Results**: After the three-week withdrawal protocol (T_1_), we observed a normalization of the baseline habituation deficit (T_0_: +0.10 ± 0.54; T_1_: −0.53 ± 0.8; *p* = 0.040) and a reduction in the amplitude for both pre-HFO (*p* < 0.009) and post-HFO (*p* = 0.042), with values comparable to those of the HVs. However, no effects were observed on the dLI (*p* = 0.141). **Conclusions**: Our findings showed that withdrawal from overused drugs could affect the increased excitability of the non-painful somatosensory system in patients with CM-MOH, reducing the level of sensitization at both the thalamocortical and the cortical levels.

## 1. Introduction

Medication-overuse headache (MOH) is a secondary chronic headache caused by regular medication overuse in patients with a pre-existing primary headache disorder [1]. The incidence of MOH is estimated at around 0.72 cases per 1000 person-years [2], with an adult prevalence in the general population of 1–2% [3]. MOH has a substantial negative impact on patients’ functioning and quality of life [4], with high related costs [5]. Individuals with MOH have higher healthcare resource utilization and costs compared to individuals without MOH [5]. Therefore, understanding its pathophysiology is essential for identifying effective treatments and reducing healthcare and socioeconomic costs. 

From a neurophysiological point of view, MOH is characterized by increased cortical excitability, alteration of pain processing with sensitization, and functional reorganization of different cerebral areas and networks [6]. In animal models, these alterations were related to the chronic administration of simple analgesics or triptans [7,8,9]. 

Reducing or withdrawing from overused medication is essential to successfully treat this condition [6], and several studies have provided evidence that withdrawal is associated with clinical improvement [10,11,12]. It is estimated that approximately 50% of patients had a substantial improvement in headache frequency after the withdrawal [13,14]. Along with the reduction in monthly headache days, the withdrawal can influence and restore some of the neurophysiological dysfunctions that characterize patients with MOH [15,16,17]. Additionally, several neuroimaging studies showed morphologic and functional alterations in areas involved in pain modulation in individuals with chronic migraine with medication overuse headache, which could be reverted after successful treatment [18,19]. 

As MOH is considered a secondary headache in which the overconsumption of acute medications causes the chronification, understanding the neurophysiological effects of the withdrawal may help in revealing the mechanism of its development and can help in identifying new treatment strategies. We recently investigated the levels of thalamocortical activation and cortical lateral inhibition in patients with chronic migraine with MOH (CM-MOH). Our findings suggested that the central neuronal circuits in these patients are highly sensitized at both the thalamocortical and the cortical levels [20]. However, it is yet to be determined whether these alterations could be reversed by medication withdrawal (i.e., brain state) or whether they represent a brain trait in individuals predisposed to MOH. It is known that individuals with CM-MOH are characterized by other brain traits that can predispose them to medication overuse [21]. Indeed, brain regions associated with addictive behavior as part of the mesocorticolimbic dopamine system, such as the orbitofrontal cortex and the substantia nigra/ventral tegmentum, tend to remain abnormal even after drug withdrawal [16,21,22,23,24].

In this study, we aimed to understand whether the increased thalamocortical and cortical sensitization of the non-painful somatosensory system could be influenced by withdrawal from the overused drugs. For this reason, we recorded somatosensory evoked potentials (SSEPs) in patients with CM-MOH before and after a three-week acute medication withdrawal period. 

We hypothesized that withdrawal from the overused drug could reduce the central sensitization at both the thalamocortical and the cortical levels.

## 2. Materials and Methods

### 2.1. Participants

Twenty-seven patients who attended the headache clinic of the Sapienza University of Rome (Polo Pontino, ICOT, Latina) and were diagnosed with chronic migraine with medication-overuse headache (CM-MOH), based on the diagnostic criteria of *The International Classification of Headache Disorders, third edition* (ICHD-III) [1], were invited to participate in this study. All the patients received a headache diary via email at least one month before their screening visit, which was included in the waiting list for consultation. The exclusion criteria were as follows: diagnosis of secondary headache, other type of headache than migraine as primary headache, history of other diseases including neurological disorders and neuro-ophthalmological conditions, and prophylactic migraine medications in the previous three months. Clinical and demographic characteristics were collected at the screening visit. 

For comparison, we recorded 14 healthy volunteers (HVs) with a comparable age and gender distribution (33.93 ± 10.38 years; 12 female). The inclusion criteria for the control group were no personal or familial history of migraine (first-degree relatives), no other significant medical conditions (including neurological or psychiatric disease), and no regular medication intake except for the contraceptive pill. To minimize variability due to hormonal changes, we ensured that women in the control group were recorded outside their premenstrual or menstrual periods (within ±2 days of menstruation). All the study participants underwent a neuro-ophthalmological evaluation, which included measuring intraocular pressure, best-corrected visual acuity, slit-lamp biomicroscopy, and indirect ophthalmoscopy. All the participants provided written informed consent after receiving a comprehensive description of this study. This study was approved by the local ethics review board (RIF.CE 4839, date: 28 June 2019) and conducted in accordance with the Declaration of Helsinki.

### 2.2. Withdrawal Protocol

The three-week withdrawal program was implemented with the advice to abruptly withdraw the overused medication without any prophylactic medication [11]. We chose the duration of the program according to our previous protocol [17]. The patients were allowed to use a different class from the overused drug as a rescue medication to treat a maximum of one attack per week (i.e., patients who overused NSAIDs were allowed to take triptans, and vice versa; multiple assumptions during the day were allowed if needed). Corticosteroids were not allowed as rescue medications as they have the potential to affect cortical excitability [23]. Two investigators (C.A. and F.C.) telephonically interviewed the patients weekly to assess their compliance with the withdrawal protocol. The patients who exceeded the limit of one day per week with tablet intake were considered withdrawal failures and were excluded from this study.

### 2.3. Data Acquisition

Patients with CM-MOH were recorded before (T_0_) and three weeks after (T_1_) withdrawal from their overused drugs, while the HVs were recorded once. Both groups were recorded in the same laboratory and during the same period of the day. They underwent three recording sessions of SSEPs, with each session separated by five minutes of rest. Due to the frequency of acute medication use by patients with MOH, we were unable to exclude individuals who had taken medication on the day of the recordings. However, we managed to conduct the recordings at least three hours after the last medication intake.

The SSEPs were elicited using the same procedure used in our previous work [20]. The stimulation was performed using constant-current square wave pulses (0.2 ms width, cathode positioned proximally) with the intensity set at 1.2 times the motor threshold and a repetition rate of 2.2 Hz.

Before the recordings, the ground electrode was positioned on the left wrist, while three active electrodes were placed as follows: the first over Erb’s point on the ipsilateral to the stimulus and referenced to the opposite side, the second one over the fifth cervical spinous process (Cv5), and the third electrode in correspondence to the contralateral parietal area (C3′, 2 cm posterior to C3 in the International 10–20 system). Both the second and the third electrode were referenced to Fz. The SSEP signals were recorded using a CED^TM^ power 1401 device (Cambridge Electronic Design Ltd., Cambridge, UK) and amplified with a Digitimer^TM^ (Digitimer Ltd., Welwyn Garden City Welwyn Garden City, UK) (band-pass 0.05–2500 Hz, Gain 1000). All the recordings were conducted in a well-lit room during the afternoon (between 2 p.m. and 6 p.m.). 

The procedure consisted of stimulating the median nerve at the wrist in the first session (M), stimulating the ulnar nerve at the wrist in the second session (U), and simultaneously stimulating both nerves in the third session (MU) (Figure 1). 

The participants were instructed to sit relaxed in a comfortable chair, keep their eyes open, and focus on the stimulus-induced thumb movement. We recorded 300 consecutive sweeps of 50 ms sampled at 5000 Hz for each session. Two experienced neurophysiologists (F.C. and C.A.) performed all the recordings. The recordings were analyzed offline by two independent investigators (G.C. and G.S.) using the Signal^TM^ software package version 4.10 (CED Ltd., Cambridge, UK). The Signal^TM^ artifact rejection tool was used to identify artifacts. Any signal amplitude beyond 90% of the analog-to-digital converter (ADC) range was discarded. Furthermore, all the rejections were controlled by visual inspection. 

### 2.4. Low-Frequency SSEPs (LF-SSEPs) and Cortical Lateral Inhibition

All the signals were digitally filtered within the range of 0 to 450 Hz, and 300 artifact-free evoked responses were averaged for each individual throughout the three sessions. All the SSEP components (N9, N13, N20, P25, and N33) were identified according to their latencies from traces obtained solely from median nerve stimulation. Following their identification, the peak-to-peak amplitude was computed for all the SSEP components. The 300 traces acquired from median nerve stimulation were subsequently partitioned into three separate blocks of 100 sweeps each, and the peak-to-peak amplitude of each block’s responses was quantified. Consistent with our prior research, habituation was assessed by performing linear regression on the N20-P25 amplitudes of the first and second blocks, as well as between the first and third blocks [24]. The degree of cortical lateral inhibition (dLI) was calculated using the formula: [100 − (MU/(M + U) × 100)] [20]. The symbol MU in this formula represents the amplitude of the SSEP component obtained after simultaneous stimulation of median (M) and ulnar (U) nerves, while M+U is the arithmetic sum obtained by stimulating these nerves separately.

### 2.5. SSEP High-Frequency Oscillations (HFOs)

The high-frequency oscillations within the parietal N20 LS-SSEP component were extracted using a digital zero-phase-shift band-pass filter, ranging from 450 to 750 Hz (Bartlett–Hanning window, 51 filter coefficients), applied offline to SSEPs elicited only by median nerve stimulation. 

Two components of the HFOs were visually recognized based on the pronounced reduction in amplitude and frequency: the early pre-synaptic burst and the late post-synaptic burst. In cases in which differentiation between these two components based on frequency and amplitude reduction was unfeasible, we classified the bursts preceding the N20 peak as pre-synaptic and those succeeding this peak as post-synaptic. The initial pre-synaptic burst develops during the latency period of the ascending slope of the LF-SSEP N20 component. The late post-synaptic burst develops on the declining slope of N20 and occasionally extends into the ascending slope of the N33 peak. The stimulus artifact was removed from all traces, and the latency of the negative oscillatory maximum, as well as the peak-to-peak amplitude for both bursts, was assessed. 

### 2.6. Statistical Analysis 

The sample size calculation was not based on formal statistics but on previous studies that prospectively investigated patients with CM-MOH before and after withdrawal [17,24].

We used the software Jamovy for all the analyses (The jamovi project (2024). *Jamovi* (Version 2.5), Computer Software). Retrieved from https://www.jamovi.org. To assess the normal distribution of data, each electrophysiological parameter was tested using the Shapiro–Wilk test. A paired-sample *t*-test was used to compare normally distributed data between T_0_ and T_1_; otherwise, a non-parametric Wilcoxon rank test was employed. In contrast, an independent-sample *t*-test was used to compare normally distributed data between individuals with CM-MOH and HVs; otherwise, the Mann–Whitney U test was used for non-normally distributed and the Welch’s *t*-test for normally distributed data but with violation of the assumption of equal variances (significant Levene’s test = *p* < 0.05). For all the analyses, a *p*-value < 0.05 was considered statistically significant. 

We used Pearson’s correlation coefficient and corrected for multiple comparisons to assess the correlation between significant electrophysiological data at T_1_ (N20-P25 amplitude slope between blocks 1 and 2, amplitude of pre-HFO and post-HFO), and clinical variables at T_1_ (number of headache days/3 weeks, total tablet intake, duration of the chronic phase, duration of the overuse phase, headache severity, headache duration). To compensate for the number of clinical variables, *p*-values of less than 0.0025 were considered to reflect statistical significance.

## 3. Results

### 3.1. Clinical Data

In total, 14 out of the 27 patients with CM-MOH completed the withdrawal protocol and were included in the final analysis (12 female, mean age 40.5 ± 14.37) and compared with 14 age- and gender-matched HVs (12 female, mean age 33.93 ± 10.38). The remaining thirteen patients were excluded from this study as they failed the three-week withdrawal protocol, exceeding the limit of one day per week with tablet intake. 

After the three-week withdrawal protocol, we observed a significant reduction in the number of headache days (T_0_ = 24.43 ± 5.02; T_1_ = 11.21 ± 6.6; T_0_ vs. T_1_: t = 5.25, *p* ≤ 0.001), while no differences emerged regarding the headache duration (T_0_ vs. T_1_: Z = 3.00, *p* = 1.000) and severity (VAS T_0_ vs. T_1_: Z = 8.00; *p* = 0.345). Table 1 provides the demographics and clinical features of the participants included in the analysis.

### 3.2. LF-SSEPs and Cortical Lateral Inhibition

All the participants included in the final analysis had assessable SSEP recordings. The grand average N9, N13, N20, P25, and N33 latencies and the peak-to-peak amplitudes of N9 and N13 did not differ between the HVs and individuals with CM-MOH before (T_0_) and after the withdrawal protocol (T_1_). 

In comparison with the HVs, individuals with CM-MOH showed baseline (T_0_) increased peak-to-peak amplitudes for grand average N20-P25 (t = −2.23, *p* = 0.035) and P25-N33 (U = 34.00, *p* = 0.004), and for the second and third blocks of N20-P25 averaged responses (second block: t = −2.32, *p* = 0.029; third block: U = 41.00, *p* = 0.009) (Table 2, Figure 2a). 

After the three-week withdrawal protocol (Table 3), we observed a reduction in the amplitude of the second block of responses (second block: Z = 93.00, *p* = 0.009), which returned to a value comparable with that of the HVs (second block HV vs. CM-MOH T_1_: t = −0.21, *p* = 0.834). No differences emerged regarding the first and the third block between T_0_ and T_1_ (first block T_0_ vs. T_1_: t = 0.57; *p* = 0.581; third block T_0_ vs. T_1_: t = 1.24; *p* = 0.237). 

Contrary to the HVs, the individuals with CM-MOH exhibited a progressive increase in the N20-P25 amplitude slope at T_0_, indicating deficient habituation. This increase was observed between blocks 1 and 2 (Slope 1–2: HV = −0.27 ± 0.23; CM-MOH T_0_ = 0.10 ± 0.54; t = −1.49, *p* = 0.031) and between blocks 1 and 3 (Slope 1–3: HV = −0.17 ± 0.21; CM-MOH T_0_ = 0.20 ± 0.63; U = 46.50, *p* = 0.019) (Table 2, Figure 2b,c). After the withdrawal (T_1_), the N20-P25 amplitude slope between blocks 1 and 2 showed a significant reduction (Slope 1–2: T_0_ = 0.10 ± 0.54; T_1_ = −0.53 ± 0.8; T_0_ vs. T_1_: t = 2.28, *p* = 0.040), returning within values comparable to those for the HVs (Slope 1–2: HV = −0.27 ± 0.23; CM-MOH T_1_ = −0.53 ± 0.8; U = 78.00, *p* = 0.370), while no effects were observed on the N20-P25 amplitude slope between blocks 1 and 3 (T_0_ vs. T_1_: t = 1.37, *p* = 0.195) (Table 3, Figure 3).

In contrast with our previous evidence [18], no differences emerged for dLI between HVs and individuals with CM-MOH at baseline (HV = 40.37 ± 14.89; CM-MOH T_0_ = 45.24 ± 12.81; t = −0.93, *p* = 0.362). Although individuals with CM-MOH at T_1_ had a higher dLI than the HVs (CM-MOH T_1_ = 57.35 ± 22.3; HV = 40.37 ± 14.89; t = −2.37, *p* = 0.026) (Table 4), no differences emerged between the T_0_ and T_1_ values of individuals with CM-MOH (T_0_ = 45.24 ± 12.81; T_1_ = 57.35 ± 22.3; T_0_ vs. T_1_: t = −1.57; *p* = 0.141) (Table 3).

### 3.3. HFOs

We found no differences between T_0_ and T_1_ in the latency of the negative oscillatory maximum peak of both pre-HFO (t = 0.51, *p* = 0.62) and post-HFO (t = −0.28, *p* = 0.785). In comparison with the HVs, the individuals with CM-MOH showed baseline (T_0_) increased peak-to-peak amplitudes for pre-HFO (HV = 0.07 ± 0.03; CM-MOH T_0_ = 0.11 ± 0.06; t = 18.72, *p* = 0.015) and post-HFO (HV = 0.08 ± 0.03; CM-MOH T_0_ = 0.12 ± 0.07; t = 16.59, *p* = 0.043) (Table 2, Figure 4), which both reduced (pre-HFO amplitude: T_0_ = 0.11 ± 0.06, T_1_ = 0.06 ± 0.03; T_0_ vs. T_1_: t = 3.08, *p* < 0.009; post-HFO amplitude: T_0_ = 0.12 ± 0.07, T_1_ = 0.08 ± 0.04; T_0_ vs. T_1_: t = 2.25; *p* = 0.042) (Table 3, Figure 5) and returned to normal values at T_1_ (pre-HFO amplitude: HV = 0.07 ± 0.03; CM-MOH T_1_ = 0.06 ± 0.03; U = 73.00, *p* = 0.254; post-HFO amplitude: HV = 0.08 ± 0.03; CM-MOH T_1_ = 0.08 ± 0.04; t = −0.41, *p* = 0.687) (Table 4). 

### 3.4. Correlation Analyses

No significant correlations emerged between neurophysiological parameters and the clinical data at baseline (Appendix A).

## 4. Discussion

The main aim of the present study was to investigate the influence of withdrawal from overused drugs on subcortico-cortical excitability in patients with CM-MOH to determine whether the increased central sensitization could be reverted (i.e., a brain state) or whether it is a brain trait of individuals predisposed to MOH.

We confirmed that individuals with CM-MOH had increased sensitization at both the thalamocortical (increased amplitude of pre-HFO) and the cortical levels (increased amplitudes of grand-average N20-P25, P25-N33, and post-HFO) in comparison to the HVs. In contrast, we failed to confirm our previous result of the increased degree of cortical lateral inhibition in individuals with CM-MOH [20]. Indeed, although individuals with CM-MOH at T_1_ had significantly a higher dLI than the HVs, the withdrawal did not influence cortical lateral inhibition, and no differences emerged between the baseline values of individuals with CM-MOH and HVs. We could not exclude that the lack of the baseline difference in the dLI is due to the relatively small sample size, which prevented us from confirming the increased lateral inhibition shown in our previous study with a greater sample size [20]. Another intriguing hypothesis could be that more time is needed to plastically change for some neurophysiological responses, such as the dLI or the N20-P25 amplitude slope between blocks 1 and 3.

However, our study’s main finding is the evidence of the withdrawal’s influence on the non-painful somatosensory system, which normalized the increased sensitization at both the thalamocortical and the cortical levels. Different brain regions involved in pain processing were found to have atypical structures and functions in MOH [19]. Several studies have shown that withdrawal from overused drugs can positively remodel these alterations [16,22,25,26,27,28], restoring the normal neurophysiological properties of these areas [15,17,24,29]. It was found that the facilitation of trigeminal and somatic nociceptive systems was normalized after withdrawal independently from the class of overused medication [15]. Similar findings were observed using CO_2_-laser-evoked potentials, in which the deficient habituation to pain was restored following successful treatment [24]. However, these studies associated the withdrawal with the starting of preventive treatment, including drugs that can directly influence neurophysiologic responses [30,31]. Few neurophysiological studies examined the effects of withdrawal in subjects with CM-MOH without starting a prophylactic treatment. Although after a short withdrawal period (8–10 days), Perrotta et al. showed improved spinal cord pain processing and increased antinociceptive activity of the supraspinal structures [29]. Using high-frequency (5 Hz) repetitive transcranial magnetic stimulation (rTMS), a method known to forcedly increase cortical excitability, Cortese et al. found that a 3-week drug withdrawal restored the normal short-term synaptic potentiation of the primary motor cortex [17]. 

However, no previous neurophysiology studies in individuals with MOH investigated the effects of drug withdrawal on the somatosensory non-painful system. 

In this study, we provided neurophysiological evidence that drug withdrawal can influence the non-painful somatosensory system by reducing sensitization at both the thalamocortical and the cortical levels (i.e., decrease in amplitude for both pre-HFO and post-HFO). These findings align with the functional neuroimaging and FDG-PET findings, which showed that withdrawal restored the normal function of the primary somatosensory cortex [25] and reversed the hypometabolism of the thalamus [16]. Given the normalization of these parameters after the withdrawal, it could be hypothesized that the central sensitization may be due to a primary effect of the drugs’ overconsumption rather than an indirect effect due to the increased headache frequency.

Consistent with this hypothesis, animal models of medication overuse suggested that abortive migraine drugs may play a pivotal role in developing sensitization and increased cortical excitability. The continuative administration of sumatriptan or analgesics in rodents causes central sensitization [32] and enhances the frequency and susceptibility to evoked cortical spreading depression [7,8,9]. Furthermore, medication overuse may influence the serotoninergic system [33,34], which is implicated in migraine pathogenesis [35]. Indeed, chronic paracetamol administration in rats downregulates the 5-HT2A receptor and increases the serotonin (5-HT) levels in the pons and cortex and the number of 5-HT transporters in the frontal cortices [33,34]. The serotoninergic dysfunction was previously hypothesized to contribute to thalamocortical dysrhythmia, explaining the lack of habituation and reduced thalamic activity observed in individuals with episodic migraine during the interictal period [36]. Based on this evidence, it is tempting to speculate that medication overuse may further disrupt the dysfunctional serotoninergic systems of individuals with episodic migraine, leading to increased cortical excitability and sensitization, which predisposes these patients to chronicity.

We acknowledge that our study has some limitations. One of the major limitations of our study is the relatively small sample size and the female-to-male ratio, which is not in line with the one reported in the literature [37]; these limitations could influence the generalizability of our results. Additionally, the small sample size also precluded a subanalysis of patients according to the type of overused medication. The lack of a control group with chronic migraine without medication overuse is another limitation of our study. However, it is important to note that previous evidence using the same paradigm revealed a different behavior for lateral inhibition in CM patients without medication overuse [38]. Additionally, we did not collect information about psychiatric comorbidities or personality traits, which prevented us from excluding patients with these comorbidities. This could have influenced our results as previous studies have found that these comorbidities can influence neurophysiological responses [39,40]. Another limitation of our study is that the patients were not followed-up after three or six months. As a result, we cannot completely rule out the possibility that the observed changes were transitory, and some of the neurophysiological responses (such as the dLI or the N20-P25 amplitude slope between blocks 1 and 3) could need more time to plastically change. However, we acknowledge that a longer follow-up is only possible with preventive medications. Contrary to most studies, each patient in our study underwent a pure withdrawal protocol without the concomitant use of any preventive drugs. This excludes the drugs’ influence on the neurophysiological responses and the biased reduction in the number of headache days. Finally, it is important to note that our low inclusion rate (14/27) does not fully reflect the efficacy of the withdrawal from a clinical point of view. We applied a strict threshold to the number of rescue medications allowed per week to reduce their possible influence on the neurophysiological variables. It is plausible to hypothesize that the inclusion rate would be higher with a less strict threshold for the number of rescue medications allowed.

## 5. Conclusions

In conclusion, we showed that a three-week withdrawal protocol can influence the non-painful somatosensory system in a group of patients with CM-MOH, mainly female patients, by normalizing the increased excitability at both the thalamocortical and the cortical levels. It could be hypothesized that central sensitization in CM-MOH primarily develops due to medication overuse, which may further disrupt the dysfunctional serotoninergic systems of individuals with migraine and predispose them to chronicity. Further studies with a greater sample size, with the inclusion of a control group of individuals with chronic migraine without medication overuse, and with a follow-up extension of up to 6 months after the withdrawal are needed to confirm our results. Finally, the gender differences in the neurophysiological response to the withdrawal should be investigated in future research. 

## Figures and Tables

**Figure 1 jcm-13-07491-f001:**
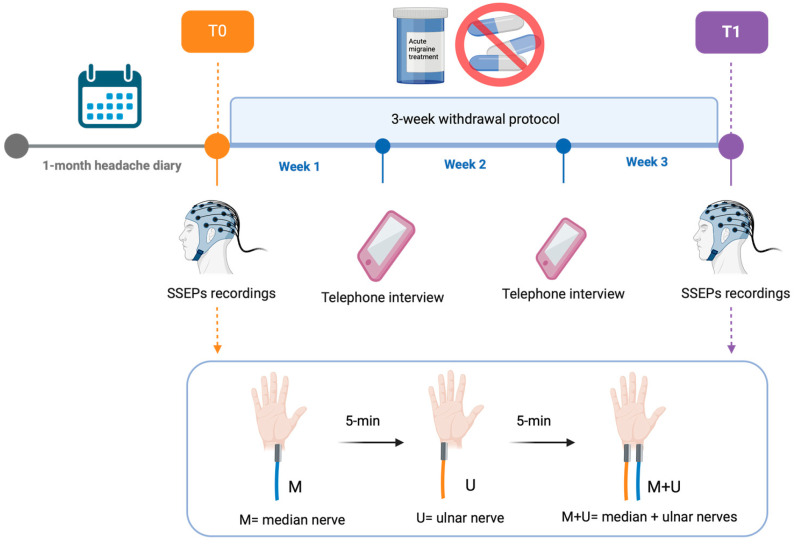
Flow chart of the study’s protocol and recording sessions. Created in BioRender. Sebastianelli, G. (2024) https://BioRender.com/a54g519.

**Figure 2 jcm-13-07491-f002:**
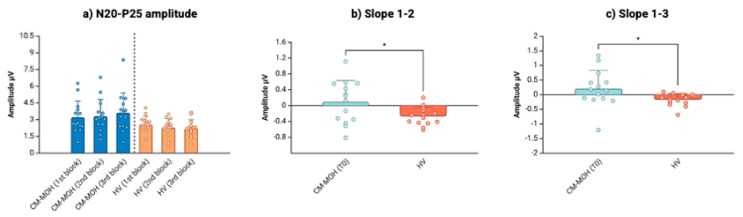
(**a**) Amplitude for the first three blocks of 100 sweeps of N20-P25 averaged responses (mean ± SD); (**b**) Slope (linear regression, mean ± SD) between the N20-P25 amplitudes of the first and second blocks and (**c**) between the first and third blocks in patients with CM-MOH at baseline (T_0_) and HVs. * = *p* < 0.05. Created on https://BioRender.com.

**Figure 3 jcm-13-07491-f003:**
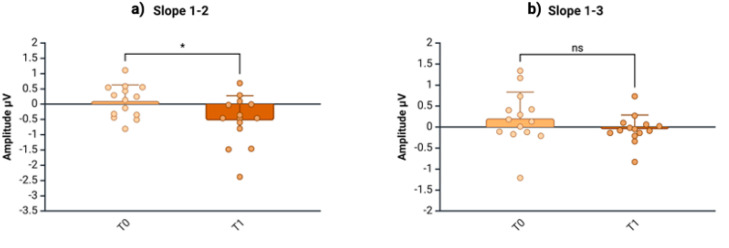
Slope (linear regression, mean ± SD) between the N20-P25 amplitudes of the first and second blocks (**a**) and between the first and third blocks (**b**) in patients with CM-MOH before (T_0_) and after the 3-week withdrawal protocol (T_1_). ns = *p* > 0.05; * = *p* < 0.05. Created in https://BioRender.com.

**Figure 4 jcm-13-07491-f004:**
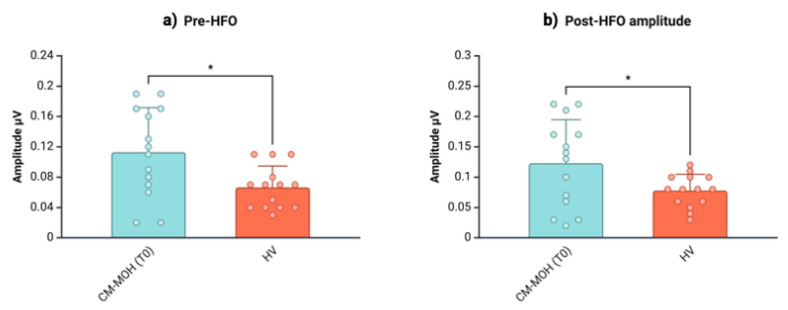
Amplitudes (mean ± SD) of pre-HFO (**a**) and post-HFO (**b**) of SSEP high-frequency oscillations (HFOs) in patients with CM-MOH at baseline (T_0_) and HVs. * = *p* < 0.05. Created on https://BioRender.com.

**Figure 5 jcm-13-07491-f005:**
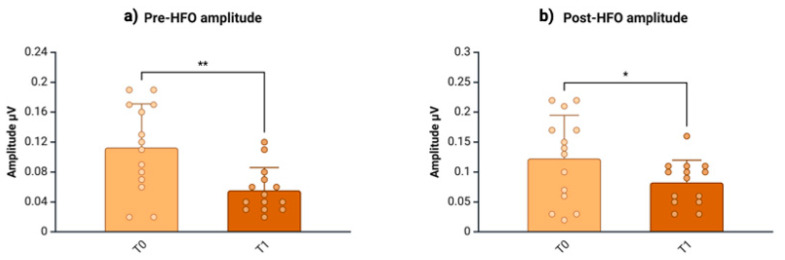
Amplitudes (mean ± SD) of pre-HFO (**a**) and post-HFO (**b**) of SSEP high-frequency oscillations (HFOs) in patients with CM-MOH before (T_0_) and after the 3-week withdrawal protocol (T_1_). * = *p* < 0.05; ** = *p* < 0.01. Created on https://BioRender.com.

**Table 1 jcm-13-07491-t001:** Demographics and clinical features.

	HV	CM-MOH		*p*-Value
Female (%)	12 (85.7%)	12 (85.7%)		
Age (years)	33.93 ± 10.38	40.5 ± 14.37		0.177
MHDs (n)		24.43 ± 5.02		
Total tablet intake/month (n)		25.57 ± 14.73		
Duration of the chronic phase (years)		18.79 ± 16.32		
Duration of the overuse (years)		16.86 ± 16.45		
Triptan overusers (n)		3 (21.4%)		
NSAID overusers (n)		8 (57.1%)		
Combination medication overusers	(n)	3 (21.4%)		
		T_0_	T_1_	*p*-Value
Total tablet intake/3 weeks (n)		17.90 ± 10.31	1.86 ± 1.03	<0.001
Number of headache days/3 weeks		17.10 ± 3.51	11.21 ± 6.6	0.021
Headache duration (h)		11.14 ± 2.18	14.79 ± 16.68	1.000 ^Z^
Headache severity (VAS 0–10)		7.61 ± 1.94	7.18 ± 2.03	0.345 ^Z^

Data expressed as mean ± standard deviation. Abbreviations: h = hours; n = number; MHDs = monthly headache days; T_0_ = baseline; T_1_ = after three-week withdrawal protocol; VAS = visual analog scale; ^Z^ = Wilcoxon rank test.

**Table 2 jcm-13-07491-t002:** Comparison of neurophysiological variables between HVs and individuals with CM-MOH before the three-week withdrawal protocol (T_0_).

	HV	CM-MOH (T_0_)	*p*-Value
N20-P25 (μv)	2.04 ± 0.78	3.04 ± 1.49	0.035
P25-N33 (μv)	0.91 ± 0.57	1.52 ± 0.62	0.004 ^U^
1st block N20-P25 (μv)	2.56 ± 0.76	3.21 ± 1.44	0.152 ^w^
2nd block N20-P25 (μv)	2.29 ± 0.79	3.33 ± 1.48	0.029
3rd block N20-P25 (μv)	2.22 ± 0.77	3.62 ± 1.78	0.009 ^U^
Slope (blocks 1–2)	−0.27 ± 0.23	0.10 ± 0.54	0.031 ^w^
Slope (blocks 1–3)	−0.17 ± 0.21	0.20 ± 0.63	0.019 ^U^
dLI	40.37 ± 14.89	45.24 ± 12.81	0.362
Pre-HFO latency (ms)	15.96 ± 2.21	16.68 ± 1.51	0.324
Pre-HFO amplitude (ms)	0.07 ± 0.03	0.11 ± 0.06	0.015 ^w^
Post-HFO latency (ms)	23.40 ± 2.86	22.69 ± 2.96	0.581 ^U^
Post-HFO amplitude (ms)	0.08 ± 0.03	0.12 ± 0.07	0.043 ^w^

Data expressed as mean ± standard deviation. Abbreviations: dLI = degree of lateral inhibition; mA = milliampere; ms = millisecond; μv = microvolt; T_0_ = baseline; T_1_ = after three-week withdrawal protocol; ^U^ = Mann–Whitney U test; ^w^ = Welch’s *t*-test.

**Table 3 jcm-13-07491-t003:** Neurophysiological variables of individuals with CM-MOH before (T_0_) and after the three-week withdrawal protocol (T_1_).

	T_0_	T_1_	*p*-Value
N20-P25 (μv)	3.04 ± 1.49	2.67 ± 1.40	0.502 ^Z^
P25-N33 (μv)	1.52 ± 0.62	1.30 ± 0.83	0.378
1st block N20-P25 (μv)	3.21 ± 1.44	3.00 ± 1.60	0.581
2nd block N20-P25 (μv)	3.33 ± 1.48	2.38 ± 1.45	0.009 ^Z^
3rd block N20-P25 (μv)	3.62 ± 1.78	2.87 ± 1.64	0.237
Slope (blocks 1–2)	0.10 ± 0.54	−0.53 ± 0.8	0.040
Slope (blocks 1–3)	0.20 ± 0.63	−0.05 ± 0.34	0.195
dLI	45.24 ± 12.81	57.35 ± 22.3	0.141
Pre-HFO latency (ms)	16.68 ± 1.51	16.39 ± 1.33	0.62
Pre-HFO amplitude (ms)	0.11 ± 0.06	0.06 ± 0.03	0.009
Post-HFO latency (ms)	22.69 ± 2.96	22.92 ± 2.39	0.785
Post-HFO amplitude (ms)	0.12 ± 0.07	0.08 ± 0.04	0.042

Data expressed as mean ± SD. Abbreviations: dLI = degree of lateral inhibition; mA = milliampere; ms = millisecond; μv = microvolt; T_0_ = baseline; T_1_ = after three-week withdrawal protocol; ^Z^ = Wilcoxon rank test.

**Table 4 jcm-13-07491-t004:** Comparison of neurophysiological variables between HVs and individuals with CM-MOH after the three-week withdrawal protocol (T_1_).

	HV	CM-MOH (T_1_)	*p*-Value
N20-P25 (μv)	2.04 ± 0.78	2.67 ± 1.40	0.158 ^w^
P25-N33 (μv)	0.91 ± 0.57	1.30 ± 0.83	0.183 ^U^
1st block N20-P25 (μv)	2.56 ± 0.76	3.00 ± 1.60	0.362 ^w^
2nd block N20-P25 (μv)	2.29 ± 0.79	2.38 ± 1.45	0.834 ^w^
3rd block N20-P25 (μv)	2.22 ± 0.77	2.87 ± 1.64	0.197 ^w^
Slope (blocks 1–2)	−0.27 ± 0.23	−0.53 ± 0.80	0.370 ^U^
Slope (blocks 1–3)	−0.17 ± 0.21	−0.05 ± 0.34	0.223 ^U^
dLI	40.37 ± 14.89	57.35 ± 22.30	0.026
Pre-HFO latency (ms)	15.96 ± 2.21	16.39 ± 1.33	0.301 ^U^
Pre-HFO amplitude (ms)	0.07 ± 0.03	0.06 ± 0.03	0.254 ^U^
Post-HFO latency (ms)	23.40 ± 2.86	22.92 ± 2.39	0.769 ^U^
Post-HFO amplitude (ms)	0.08 ± 0.03	0.08 ± 0.04	0.687

Data expressed as mean ± SD. Abbreviations: dLI = degree of lateral inhibition; mA = milliampere; ms = millisecond; μv = microvolt; T_0_ = baseline; T_1_ = after three-week withdrawal protocol; ^U^ = Mann–Whitney U test; ^w^ = Welch’s *t*-test.

## Data Availability

The data that support the findings of this study are available from the corresponding author upon reasonable request.

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
