# Peer review of "Neurophysiological Effects of Withdrawal from Acute Overused Medications in Chronic Migraine with Medication-Overuse Headache"

_jcm, 2024, doi:10.3390/jcm13237491_

Round 1

Reviewer 1 Report

Comments and Suggestions for Authors

Dear Authors,

Thanks a lot for giving me the opportunity to revise the fascinating manuscript "Neurophysiological effects of withdrawal from acute overused medications in chronic migraine with medication overuse headache" the following suggestion could improve the quality of your work.

Introduction

*       While it introduces the concept of withdrawal in chronic migraine with medication overuse headache, it could more precisely delineate the specific research gap this study aims to fill.

*       The introduction would benefit from a clearer statement of the study's objectives and hypotheses. Explicitly defining these early on would enhance the reader's understanding of the study's direction.

 * A more comprensive overview of neurophysiological studies of migraine pathophysiology could support your work, please take in to consideration the following articles:

Deodato M, Granato A, Martini M, Buoite Stella A, Galmonte A, Murena L, Manganotti P. Neurophysiological and Clinical Outcomes in Episodic Migraine Without Aura: A Cross-Sectional Study. J Clin Neurophysiol. 2024 May 1;41(4):388-395. doi: 10.1097/WNP.0000000000001055. PMID: 37934069.

Lai TH, Wang SJ. Neuroimaging Findings in Patients with Medication Overuse Headache. Curr Pain Headache Rep.  2018 Jan 16;22(1):1. doi: 10.1007/s11916-018-0661-0. PMID: 29340793.

-Chen W, Li H, Hou X, Jia X. Gray matter alteration in medication overuse headache: a coordinates-based activation estimation meta-analysis. Brain Imaging Behav. 2022 oct 16(5):2307-2319. doi: 10.1007/s11682-022-00634-9. Epub 2022 Feb 10. PMID: 35143020; PMCID: PMC9581858.

DISCUSSION:
*       While some limitations are acknowledged, a more comprehensive discussion of the study's limitations, including the small sample size would provide a more balanced view.

 *       The discussion would benefit from more detailed suggestions for future research, including the exploration of long-term effects through extended follow-up periods.

*       Expanding on the potential mechanisms underlying the observed benefits could deepen understanding and suggest avenues for further investigation.

 *       More thorough consideration of the study's generalizability to different migraine populations and settings would help readers assess the findings' applicability.

Reviewer 2 Report

Comments and Suggestions for Authors

This study by Sebastianelli et al. investigated if neurophysiological abnormalities, including increased cortical excitability with sensitization, that are observed in chronic migraine with medication overuse headache (CM-MOH) could be restored by overuse drug withdrawal. The authors sought to utilize SSEPs using three recording sessions before and after a 3-week withdrawal protocol. The results, 3 weeks after withdrawal compared to before, showed a restoration of the baseline habituation deficit and a reduction in pre-HFO and post-HFO to HV levels, while with no effects on the degree of lateral inhibition. The data interpretations were mostly appropriate based on the obtained results, and the study provides further evidence of the effect of medication withdrawal on reducing sensitization, in addition to previously published fMRI and FDG-PET imaging studies. Nevertheless, there are intrinsic limitations in the study design that undermine the confidence in some data interpretation and the significance of the study in general. My detailed comments are below:

Main concerns/comments:

1.     Based on the current study design, conclusions comparing CM-MOH patients and HVs should be limited given few controlled factors between the two groups. For example: line 294-296. HVs are only similar to CM-MOH group in age and gender distribution. In order to confidently use “restoration” or “normalization”, ideally the baseline values of CM-MOH should be obtained even before medication overuse started (but it was often not possible), so it’s critical to control the factors as much as possible in CM-MOH vs. HV groups that could influence the results. As the authors already recognized in the manuscript, other factors such as psychiatric comorbidities or personality traits were not known.           

2.     Since the vast majority of the participants included in the analysis are female (12/14 in both CM-MOH and HV groups), the conclusion should highlight this point and may not just state “patients” (line 369). It would be worth it to analyze the data only for female patients, and to discuss any potential difference between females and males in restoring the increased excitability.

3.     Results 3.4 correlation analysis: “No significant correlations emerged between neurophysiological parameters and the clinical data” It will still be useful to show the relevant plots with their respective correlation coefficients and p-values with the neurophysiological parameters and clinical variables in the supplementary materials.

4.     It would be great to include the reasoning why a three-week acute medication withdrawal period was chosen.

5.     The reviewer suggests the authors to include a discussion on the reasons for no effects on degree of lateral inhibition.

Some minor points:

1.     Please include individual data points in all the bar graphs to show the distribution of the data.  

2.     There are two versions of CM-MOH full names used throughout the manuscript, “chronic migraine with medication overuse headache” and “chronic migraine and medication overuse headache”. It would be better to use one and be consistent.

3.     Line 63: likely a typo to state “brain state” in the brackets.

Reviewer 3 Report

Comments and Suggestions for Authors

I am reviewing the paper "Neurophysiological Effects of Withdrawal from Acute Overused Medications in Chronic Migraine with Medication Overuse Headache." In this article, the authors investigate the effects of drug withdrawal on cortical excitability in a population of patients with CM and MOH.

The neurophysiological procedures were thoroughly conducted and appropriately aligned with the authors' aims. The paper is clear and well-written in all its sections. However, in my opinion, some aspects of the study design could be improved. Firstly, it would have been optimal to divide the MOH population based on the type of acute medication taken (e.g. NSAIDs, triptans, or others), considering that different classes of medication have distinct effects on cortical excitability, as reported in the literature. Another potential limitation is the mismatch in observation periods: while the baseline observation lasted one month, the post-intervention timepoint was set at just three weeks. This discrepancy could potentially influence clinical variables (monthly headache days and monthly days of acute medication intake). Additionally, the authors noted in their limitations that the study did not include a cohort with CM without MOH, despite previous evidence suggesting differences in cortical excitability between CM and MOH. Evaluating the impact of drug withdrawal on these differences would have been valuable, and in my opinion it would be interesting to add a comment on the issue. Finally, the low inclusion rate (14 out of 27 patients) is another point worth commenting on.

In conclusion, the paper is acceptable for publication after including a brief discussion on the limitations of the study design. However, the study’s impact could be significantly enhanced by increasing the sample size and refining the study design. I recommend the authors consider these improvements for a future publication.

Round 2

Reviewer 2 Report

Comments and Suggestions for Authors

No further comments to authors.